# Genome-wide association identifies novel ROP risk loci in a multiethnic cohort

Xiaohui Li[1,29], Leah A. Owen [2,3,4,5,29✉], Kent D. Taylor[1], Susan Ostmo[6], Yii-Der Ida Chen [1], Aaron S. Coyner [6], Kemal Sonmez[6], M. Elizabeth Hartnett[7], Xiuqing Guo [1], Eli Ipp [8], Kathryn Roll[1], Pauline Genter[8], R. V. Paul Chan [9], Margaret M. DeAngelis[1,2,3,5,10,11,12], Michael F. Chiang [13,14], J. Peter Campbell [6✉], Jerome I. Rotter [1✉] & on behalf of the i-ROP Consortium*

We conducted a genome-wide association study (GWAS) in a multiethnic cohort of 920 at-risk infants for retinopathy of prematurity (ROP), a major cause of childhood blindness, identifying 1 locus at genome-wide significance level ($p < 5 \times 10^{-8}$) and 9 with significance of $p < 5 \times 10^{-6}$ for ROP $\geq$ stage 3. The most significant locus, rs2058019, reached genome-wide significance within the full multiethnic cohort ($p = 4.96 \times 10^{-9}$); Hispanic and European Ancestry infants driving the association. The lead single nucleotide polymorphism (SNP) falls in an intronic region within the Glioma-associated oncogene family zinc finger 3 (*GLI3*) gene. Relevance for *GLI3* and other top-associated genes to human ocular disease was substantiated through *in-silico* extension analyses, genetic risk score analysis and expression profiling in human donor eye tissues. Thus, we identify a novel locus at *GLI3* with relevance to retinal biology, supporting genetic susceptibilities for ROP risk with possible variability by race and ethnicity.

[1] Institute for Translational Genomics and Population Sciences, The Lundquist Institute for Biomedical Innovation; Department of Pediatrics, Harbor-UCLA Medical Center, Torrance, CA, USA. [2] Department of Ophthalmology and Visual Sciences, University of Utah, Salt Lake City, UT, USA. [3] Department of Population Health Sciences, University of Utah, Salt Lake City, UT, USA. [4] Department of Obstetrics and Gynecology, University of Utah, Salt Lake City, UT, USA. [5] Department of Ophthalmology, University at Buffalo the State University of New York, Buffalo, NY, USA. [6] Casey Eye Institute, Oregon Health & Science University, Portland, OR, USA. [7] Byers Eye Institute at Stanford University, Palo Alto, CA, USA. [8] Division of Endocrinology and Metabolism, Department of Medicine, The Lundquist Institute for Biomedical Innovation at Harbor-UCLA Medical Center, Torrance, CA, USA. [9] Department of Ophthalmology and Visual Sciences, Illinois Eye and Ear Infirmary, University of Illinois at Chicago, Chicago, IL, USA. [10] Department of Biochemistry; Jacobs School of Medicine and Biomedical Sciences, University at Buffalo/State University of New York (SUNY), Buffalo, NY, USA. [11] Department of Neuroscience; Jacobs School of Medicine and Biomedical Sciences, University at Buffalo/State University of New York (SUNY), Buffalo, NY, USA. [12] Department of Genetics, Jacobs School of Medicine and Biomedical Sciences, University at Buffalo/State University of New York (SUNY), Buffalo, NY, USA. [13] National Eye Institute, National Institutes of Health, Bethesda, MD, USA. [14] National Library of Medicine, National Institutes of Health, Bethesda, MD, USA. [29] These authors contributed equally: Xiaohui Li, Leah A. Owen. *A list of authors and their affiliations appears at the end of the paper. ✉email: Leah.owen@hsc.utah.edu; campbelp@ohsu.edu; jrotter@lundquist.org

Retinopathy of prematurity (ROP) is a retinal vascular disease that affects premature infants and is a leading cause of childhood blindness worldwide[1–3]. Birth weight (BW) less than 1500 g, prematurity less than 32 weeks gestational age (GA), and post-natal oxygen exposure confer independent ROP risk[4,5]. However, this understanding of risk and pathogenesis is incomplete as only approximately 50–65% of infants with these risk factors will develop ROP disease[4,6,7]. For example, phenotypic extremes that do not conform to current risk profiles exist; preterm infants born at large birth weights and/or older gestational ages may develop treatment-warranted ROP whereas those within the risk profiles of GA and BW do not always develop treatment-warranted ROP[8]. The current stratification also does not address prevention; although post-natal oxygen exposure is the most modifiable risk, limiting oxygen increases infant mortality[9–11]. As a result, current screening lacks specificity and interventions are unable to modify preclinical ROP risk, instead targeting ROP when retinal disease is present and there is significant clinical risk of blindness[6,12]. Improved understanding of risk will allow better detection and treatment of infants with greatest risk for severe ROP and may provide novel insights into disease patho-mechanisms in order to prevent disease.

ROP risk is multifactorial and although genetic risk is not fully elucidated, there is a growing body of evidence supporting a genetic basis for ROP, including twin studies[13–15]. To date, no GWAS has been published for ROP; case control and whole exome candidate gene approaches have reported significant risk associations between single nucleotide polymorphisms (SNP) in *brain-derived neurotrophic factor* (*BDNF*; rs7934165 and rs2049046), *thrombospondin type-1 domain-containing protein 4* (*THSD4*), *TNF* -308G/A polymorphism and *angiotensin 1 converting enzyme* insertion deletion *(ACE* ID) polymorphism[8,16–18] (Supplementary Table 1). By contrast, recent candidate work demonstrated significant protective associations between the *BDNF* SNP rs7929344 and ROP and severe ROP risk associations between SNPs within *VEGFA*, *NOS3*, and *EPAS1*[19]. While no SNP has been shown to reach genome-wide significance, meta-analysis has substantiated some associations[17] while also beginning to identify pathobiology underlying observed differences in ROP risk relative to race; an example being recent work demonstrating association of the *VEGFA* + 405 G > C polymorphism association with ROP risk in European Ancestry populations[20]. This observation aligns well with clinical observations showing racial differences in risk, including greater ROP severity and vision loss in white as compared to black preterm infants with equivalent BW and GA ROP risk[21,22]. Further, clinical progression has been shown to vary in Hispanic compared to white non-Hispanic populations[23].

Taken together, our current understanding of ROP risk is incomplete, which impairs our ability to predict infants at greatest disease risk or identify pathobiology allowing for ROP prevention. While ROP risk is multifactorial, evidence supports genetic contributions to risk and protection. Herein we present a GWAS analysis based within the iROP consortium, including collaborators from 14 academic institutions throughout the world,

demonstrating a novel ROP risk association with the SNP *GLI3 rs2058019* reaching genome-wide significance ($p = 4.90E-09$) within our multiethnic cohort. We further demonstrate broader applicability for this SNP and *GLI3* genetic variation with other forms of retinal disease characterized by pre-retinal neovascularization, namely diabetic retinopathy. Finally, we identify additional SNPs with an association significance of $\leq 5 \times 10^{-6}$ with severe ROP disease, demonstrate cross-significance with previously identified ROP-associated SNPs, and further demonstrate relevance for *GLI3* and genes correlating with top SNPs within the human eye, finding expression in both human donor neurosensory retina as well as retinal pigment epithelium (RPE).

## Results

**iROP GWAS Multiethnic Cohort Characteristics**. The iROP cohort consists of 2187 preterm infants born at gestational ages less than 32 weeks and birth weights smaller than 1250 g, placing them at risk to develop ROP. The iROP database was mined for infants with both biologic and phenotypic data for subsequent genotype-phenotype analysis. To identify genetic susceptibility regions for ROP, we conducted a genome-wide association study (GWAS) using the Illumina Infinium Global Screening Array with information from these infants. As published by our group, ROP phenotype was rigorously determined by a team of ophthalmologists and image graders with clinical expertise in ROP and phenotype assigned by consensus of 3 or more graders[8,24–29]. This subset of iROP patients represents multiple ethnic and racial groups, including 44.5% Hispanic, 35.4% White, 12.1% African American, and 8% unidentified race individuals as noted in Table 1. Further, all ROP phenotypes are represented, including 197 with ROP Stages 3, 4, or 5. (Table 1). The distribution of ROP disease, ethnicity, and race was similar between the infants with and without genetic data (Supplementary Table 2).

**A number of SNPs demonstrate significant associations with severe ROP**. GWAS analysis was performed on DNA extracted from all 920 cohort infants to identify genetic susceptibility regions for ROP severity. Analysis was done using a case control approach designed to determine variation associated with severe ROP. Cases included infants with stage 3 ROP disease or greater and Control infants included those with stage 2 or less severe disease. As depicted in Fig. 1, which visualizes all *p*-values of SNP associations for the 22 autosomal chromosomes plotted relative to chromosome location and significance, a number of SNP associations were identified. The Y axis represents the -log transformed *p*-value (red line representing GWAS significance level) and X axis is in genomic ordered by chromosomes and positions. SNPs reaching genome-wide significance, defined as $p \leq 5 \times 10^{-8}$, are visualized above the red line.

**GWAS identifies a novel ROP-associated SNP with genome-wide significance**. As noted in Table 2, we identified one genome-wide significant (GWS) locus ($p \leq 5 \times 10^{-8}$) and 9 with significance $\leq 5 \times 10^{-6}$ associated with ROP severity, defined by

## Table 1 iROP infants included in GWAS analysis.

| Race/Ethnicity | No ROP | Stage 1 | Stage 2 | Stage 3 | Stage 4 | Stage 5 | Total |
|---|---|---|---|---|---|---|---|
| European Ancestry | 129 | 57 | 72 | 58 | 10 | 0 | 326 |
| African American | 60 | 17 | 20 | 13 | 0 | 1 | 111 |
| Hispanic | 128 | 52 | 127 | 65 | 23 | 15 | 410 |
| Other | 34 | 11 | 16 | 11 | 0 | 1 | 73 |
| Total | 351 | 137 | 235 | 147 | 33 | 17 | 920 |
| % | (38.2%) | (14.9%) | (25.5%) | (16%) | (3.6%) | (1.8%) | |

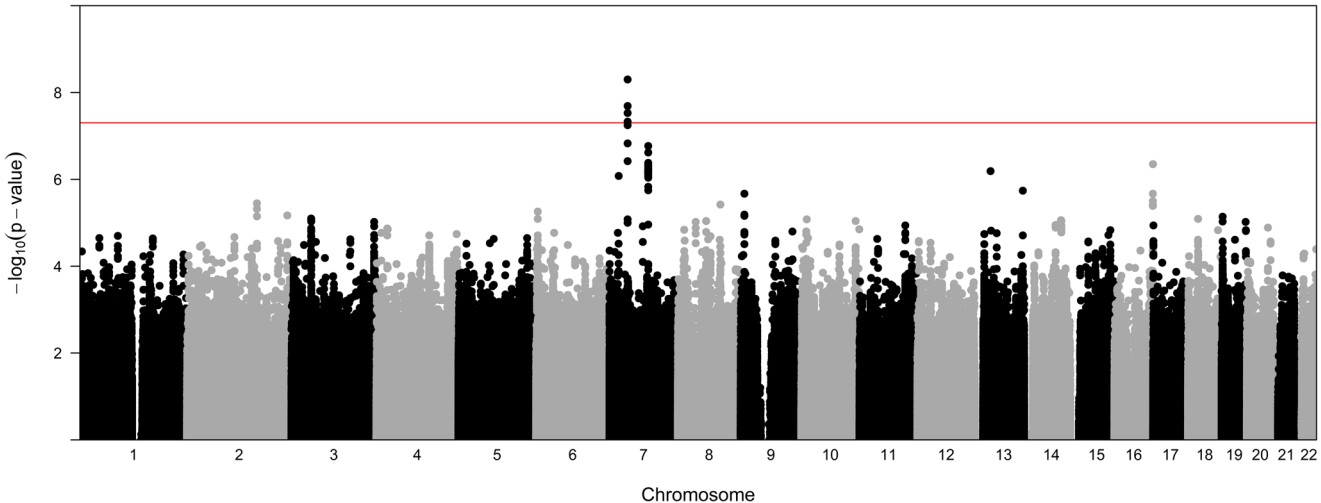

**Fig. 1 Manhattan plot of genome-wide association with ROP severity (stage 3 or greater).** Data were analyzed for all populations combined. The red line on indicates genome-wide significance ($p \leq 5 \times 10^{-8}$) with points above this line indicating SNPs with GWS.

**Table 2 Association variants identified for all ROP samples with $p < 5 \times 10^{-6}$.**

| SNP | Chr | pos | Effect Allele | AF | rsq | OR (95% CI) | p-value | p-value adj for BW & GA | Gene | eQTL p-value |
|---|---|---|---|---|---|---|---|---|---|---|
| rs2058019 | 7 | 42226712 | T | 0.06 | 0.96 | 1.27 (1.17,1.38) | 5.25E-08 | 4.96E-09* | *GLI3* | NA |
| rs11563856 | 7 | 90141855 | G | 0.17 | 0.91 | 1.14 (1.09,1.20) | 5.84E-08 | 1.71E-07 | *CLDN12* | >0.05 |
| rs62052253 | 16 | 90048889 | T | 0.22 | 0.81 | 1.13 (1.08,1.19) | 9.67E-07 | 4.50E-07 | *DEF8, TUBB3* | 0.00027 |
| rs61948265 | 13 | 36559030 | A | 0.14 | 0.68 | 1.14 (1.08,1.20) | 7.90E-06 | 6.50E-07 | *DCLK1* | 0.02 |
| rs141411503 | 7 | 21465569 | T | 0.06 | 0.89 | 1.21 (1.12,1.30) | 3.51E-05 | 8.38E-07 | *SP4* | 0.015 |
| rs78971944 | 13 | 111811973 | G | 0.09 | 0.87 | 1.17 (1.10,1.24) | 8.70E-06 | 1.83E-06 | ***ARHGEF7*** | >0.05 |
| rs9644892 | 9 | 8800078 | G | 0.37 | 0.93 | 1.10 (1.06,1.14) | 1.40E-06 | 2.14E-06 | *PTPRD* | 0.0077 |
| rs72870405 | 2 | 162937083 | A | 0.13 | 0.96 | 1.14 (1.08,1.20) | 1.13E-06 | 3.56E-06 | ***DPP4*** | >0.05 |
| rs2306129 | 8 | 99105726 | C | 0.39 | 0.84 | 1.09 (1.05,1.13) | 6.04E-05 | 3.77E-06 | *ERICH5, RPL30, RIDA* | 1.9e-21 |
| rs7751076 | 6 | 3980472 | A | 0.09 | 0.93 | 1.17 (1.09,1.25) | 9.37E-04 | 5.52E-06 | *PRPF4B* | >0.05 |

Gene names in bold text indicate a prior association with neovascular retinal disease.

stage 3 disease or greater. The position, effect allele and frequency, odds ratio, *p*-value and nearest gene for these variants is delineated in Table 2. SNPs demonstrating the lowest *p*-value in each locus/region were selected and the bioinformatically determined nearest gene was used. Under the additive genetic model, the ROP risk corresponds to the number of copies of the effect allele when the odds ratio is greater than 1. As noted, SNPs rs2058019 (*GLI3* gene) and rs11563856 (*CLDN12* gene) both reach genome-wide significance, although only rs2058019 remains significant at this level when controlling for the potentially confounding effects of BW and GA. In two of the loci with significance $\leq 5 \times 10^{-6}$, the nearest gene has a prior association with ROP disease in either humans or animal models as noted by bolded text. Finally, we performed an in-silico analysis to determine if each listed SNP had a described expression quantitative trait loci (eQTL) association in the publicly available EyeGEx[30] database. We identified a number of established regulatory associations for our top SNPs as detailed in Table 2 within the retina, further supporting relevance to the ocular microenvironment.

**rs2058019 contributes to genetic susceptibility for ROP in diverse populations.** Given established differences in clinical ROP risk based on race and ethnicity[21,22], we analyzed the association of the lead *GLI3* SNP, rs2058019, stratified by these variables. As seen in Table 3, the association between rs2058019

and ROP disease is most significant when considering all patients, although odds ratios and allele frequencies between cases and controls demonstrate that the degree and direction of association remain consistent for European Ancestry and Hispanic subjects. Interestingly, the African American subjects do not appear to be contributing to this association. While this latter finding does not reach significance in this cohort and thus requires validation within a larger population, we sought to further investigate SNP associations within the AA population as an exploratory analysis. As noted in Supplementary Data 1, we did not find evidence for significant differences in minor allele frequency (MAF) or direction of effect within racial or ethnic populations for other top SNPs. Thus, to further determine race and ethnicity-based differences in SNP associations with ROP severity, we performed our GWAS analysis separately for each racial and ethnic group. As pictured in Supplementary Fig. 1, which visually represents SNP associations for each population, racial and ethnic differences are present. As seen in Supplementary Table 3, we find several SNP associations which are racially or ethnically specific. Notably, rs9978278 (*CLDN14* gene) and rs74048122 reach genome-wide significance, which remains significant after correction for BW and GA. Although the number of observations is decreased when analyzing by race and ethnicity, these findings suggest possible differences in genetic risk for ROP by population ancestry, which requires further study within a larger and sufficiently powered multiethnic cohort.

**Table 3 Lead SNP rs2058019 demonstrates association with severe ROP disease in Hispanic and European Ancestry populations but not in African American populations.**

| Ethnicity/Race | Effect allele | AF: Case | AF: Control | AF: All | OR | P |
|---|---|---|---|---|---|---|
| HA | T | 0.22 | 0.08 | 0.11 | 1.26 | 2.95E-06 |
| EA | T | 0.04 | 0.01 | 0.01 | 1.61 | 4.0E-04 |
| AA | T | 0 | 0.02 | 0.01 | 0.87 | 0.49 |
| All | T | 0.14 | 0.04 | 0.06 | 1.27 | 4.96E-09 |

*HA* Hispanic Americans, *EA* European Ancestry, *AA* African Americans, *All* all subjects combined, *AF* allele frequency of effect allele.

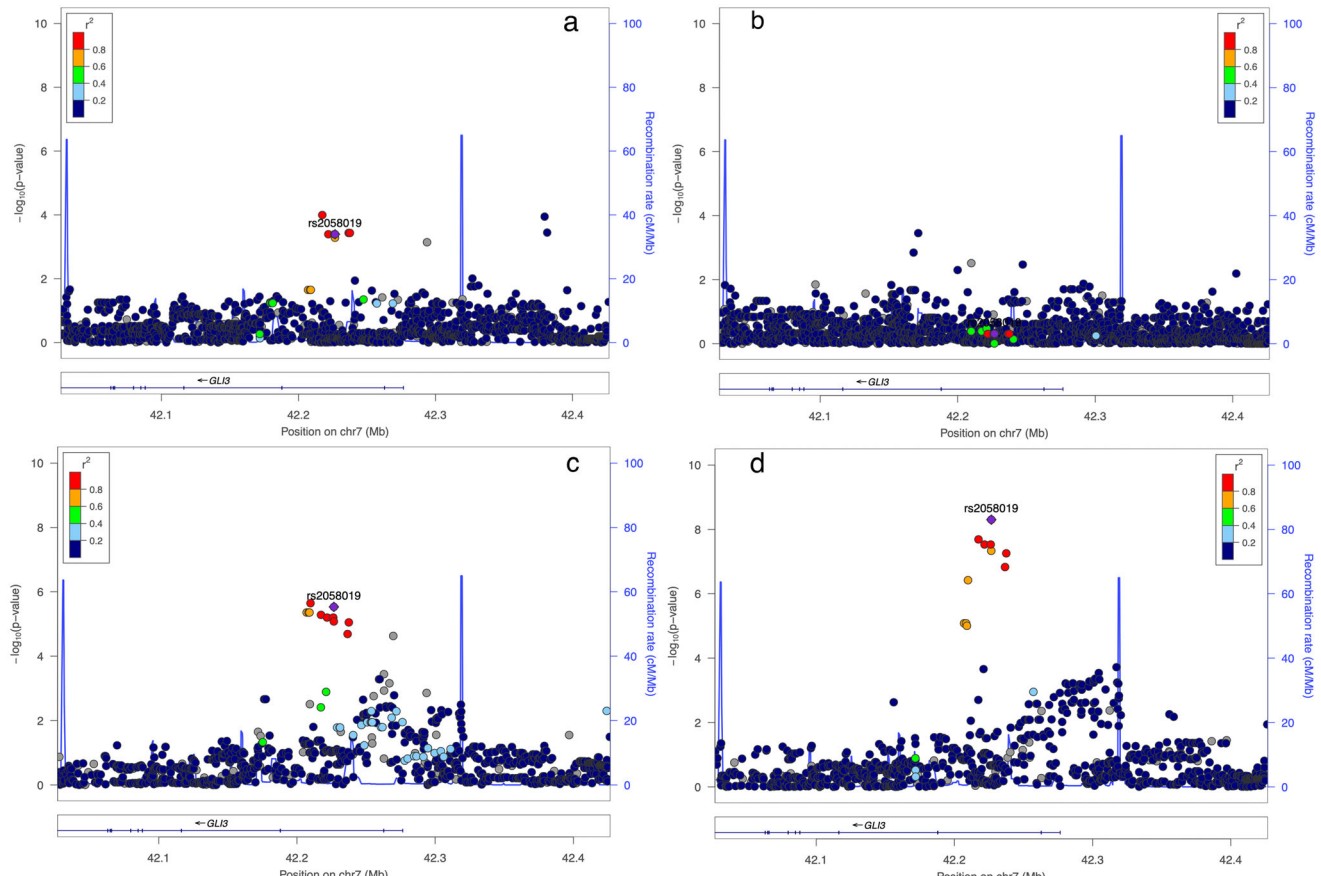

**Fig. 2 Regional association plots of top genome-wide significant SNP rs2058019 (*GLI3* gene) among each ethnic group.** Depicted in (a–d), are SNP associations for ROP severity within the *GLI3* gene for each racial or ethnic group [**a** European Ancestry; **b** African American, **c** Hispanic, and **d** all combined samples].

**Multiple SNPs are identified regionally related to *GLI3*.** To determine the regional significance for the *GLI3* pan-locus, we plotted all SNP associations within 100 Kb of the *GLI3* gene for all patients as well as for racial and ethnic subsets. As pictured in Fig. 2, a regional plot with the Y axis depicting the -log transformed *p*-value and the X axis showing chromosomal position, we illustrate the *GLI3* gene regional association for all ethnic and racial groups with ROP severity. The lead SNP rs2058019 was identified as a top-associated SNP within the pan-locus for the European Ancestry and Hispanic groups and both the best and sentinel SNP when all patients were considered together. Significant regional SNPs for each group are delineated fully in Supplementary Table 4. Also pictured, we determined the linkage disequilibrium (LD) for each SNP, defined by the *r*² using the included color scheme. LD was determined between SNPs and genes involved/close to SNPs identified.

**Top-gene loci demonstrate cross-significance for pre-retinal neovascular pathobiology within diabetic retinopathy and ROP.** Our study design, which identifies SNPs associated with ROP disease stage 3 or greater, enriches our findings for genetic variation associated with pre-retinal neovascular pathobiology. Diabetic retinopathy (DR) is another form of retinal disease characterized by pre-retinal neovascularization. Therefore, to validate the significance of *GLI3* genetic variation to pre-retinal neovascular disease, we performed an extension analysis of top-associated iROP SNPs ($p \le 5 \times 10^{-6}$) to SNPs associated with DR within the Genetics of Latino Diabetic Retinopathy (GOLDR) GWAS dataset, which consists of Hispanic patients with diabetic retinopathy (DR)[31,32]. As noted in Table 4, we found cross-significance for a number of individual SNPs with DR in GOLDR and with ROP severity in our iROP GWAS cohort. This included SNPs within *GLI3, DCLK1, SP4, PTPRD, RPL30* and *RIDA*.

**Table 4 Cross-significance of top SNPs for retinal pathology characterized by pre-retinal neovascular disease including ROP and diabetic retinopathy (GOLDR).**

| Gene | SNP | Chr | position | GOLDR | | ROP | | | LD relative to bolded SNP ($r^2$) |
|------|-----|-----|----------|-------|---|-----|---|---|------|
| | | | | odds ratio | P-value | Odds ratio | P-value | P-value Hispanic | |
| *GLI3* | rs74527981 | 7 | 42205730 | 2.22 | 9.3E-04 | Na | Na | 0.068 | 0.0006 |
| | rs17172024 | 7 | 42210112 | 2.22 | 9.3E-04 | Na | Na | 0.15 | 0.0006 |
| | **rs2058019** | 7 | 42187113 | ns | Ns | 1.27 | 4.90E-09 | 2.95E-06 | - |
| *CLDN12* | No loci associated with DR for whole region | | | | | | | | |
| *DEF8, TUBB3* | No loci associated with DR for whole region | | | | | | | | |
| *DCLK1* | rs35596426 | 13 | 36452952 | 5.18 | 4.3E-04 | Na | Na | Na | 0.007 |
| | rs9576059 | 13 | 36452266 | 4.83 | 5.9E-04 | Na | Na | Na | 0.007 |
| | **rs61948265** | 13 | 35984893 | ns | Ns | 1.14 | 6.50E-07 | 4.64E-05 | - |
| *SP4* | rs11770747 | 7 | 21552122 | 1.96 | 7.6E-04 | Ns | Ns | 0.20 | 0.02 |
| | rs78539005 | 7 | 21547126 | 2.51 | 8.5E-04 | 1.09 | 0.03 | 0.18 | 0.002 |
| | **rs141411503** | 7 | 21425951 | ns | Ns | 1.21 | 8.38E-07 | 3.55E-04 | - |
| *ARHGEF7* | No loci associated with DR for whole region | | | | | | | | |
| *PTPRD* | rs79715438 | 9 | 8809579 | 3.48 | 5.0E-04 | Na | Na | Na | 0.02 |
| | rs75240118 | 9 | 8809921 | 3.48 | 5.0E-04 | Na | Na | Na | 0.02 |
| | rs80041124 | 9 | 8810631 | 3.48 | 5.0E-04 | Na | Na | Na | 0.02 |
| | rs76777639 | 9 | 8488087 | 3.30 | 9.6E-04 | Na | Na | Na | 0.0001 |
| | rs529032346 | 9 | 8499029 | 3.30 | 9.6E-04 | Na | Na | Na | 0 |
| | **rs9644892** | 9 | 8800078 | ns | Ns | 1.1 | 2.14E-06 | 2.86E-05 | - |
| *DPP4* | No loci associated with DR for whole region | | | | | | | | |
| *RPL30, RIDA* | rs2514337 | 8 | 98113613 | 0.49 | 5.4E-04 | 0.94 | 4.4E-04 | 0.22 | 0.33 |
| | rs2447504 | 8 | 98113678 | 0.49 | 5.4E-04 | 0.94 | 4.4E-04 | 0.22 | 0.33 |
| | **rs2306129** | 8 | 98093498 | 0.66 | 0.04 | 1.09 | 3.77E-06 | 0.0078 | - |
| *PRPF4B* | No loci associated with DR for whole region | | | | | | | | |

Bolded SNP identifiers indicate the SNP used for comparison for the LD calculation.
*ns* not significant, *na* association tests not applicable for SNPs with MAF less than 0.05, *LD* linkage disequilibrium.

**Table 5 Association tests using genetic risk score (GRS) method.**

| GRS | | ROP (iROP) | | | |
|-----|------|------------|---|---|---|
| | Beta | $r^2$ (GRS only) | $r^2$ (Full model) | | P |
| SNP Score A (33 SNPs) | 3.98 | 0.25 | 0.59 | | <0.0001 |
| SNP Score B (177 SNPs) | 1.97 | 0.50 | 0.83 | | <0.0001 |
| *GLI3* only | 3.24 | 0.02 | 0.35 | | <0.0001 |

We also sought to identify the extent to which top-gene loci demonstrated cross-significance within the iROP and GOLDR datasets using linkage disequilibrium. As both the GOLDR and iROP cohorts have large Hispanic populations, this analysis was performed for Hispanic individuals. This ensures greatest rigor particularly as we found that the minor allele frequency (MAF) differed within the European Ancestry and African American populations, showing LD was not the same for all populations. As depicted in Table 4, the individual SNPs were not fully shared between the cohorts; however, we found significant linkage disequilibrium for loci within our top genes. Taken together, this increases the strength of association for multiple identified loci with the common underlying pathophysiology, pre-retinal neovascularization, and substantiates the importance of future work investigating potential differences by race and ethnicity.

**ROP Genetic Risk Score (GRS) demonstrates increased significance over single SNP associations.** Joint estimation of SNP effects has been suggested as a superior approach to improve SNP-based disease prediction and risk stratification, including in eye disease by our group[33–38]. We calculated GRSs (Score A, Score B) incorporating SNPs based on the SNP selection by LASSO method to determine if combined SNP profiles were more highly associated with ROP severity[39]. We adopted an ordinal approach for this analysis with ROP severity represented as a categorial variable denoting stage including none, stage 1, stage 2, etc and excluded SNPs demonstrating a low minor allele frequency (<0.05) for greatest rigor. Table 5 compares the association results between different GRSs. Score A consists of 33 top ROP-associated SNPs (including the top *GLI3* associated SNPs) and Score B consists of 177 top ROP-associated SNPs. As depicted in Table 5, each GRS was significantly associated with ROP severity. To determine the proportion of variation within the phenotype which can be explained using each risk score, and therefore the breadth of ROP phenotype that is associated with each GRS, we analyzed adjusted R-squared. As noted, score B accounted for a larger adjusted R-squared than the top SNP alone (0.50 versus 0.02), suggesting that the additional SNPs in this risk score together account for more variance in ROP phenotype/ severity than the *GLI3* SNP alone. As we report the only ROP GWAS study to date, and thus we could not compare to an independent dataset, these data indicate the relative effects rather than absolute estimates. Taken together, the GRS demonstrate stronger associations with all ROP phenotypes indicating there is likely greater genetic variation associated with ROP and additional studies are needed to determine the full scope of association and identify the responsible variants.

**Significant retinal vascular SNPs demonstrate relevance to ROP risk.** To determine the relevance of our dataset to vascular pathology more broadly, we analyzed the association of significant retinal vascular SNPs, identified in the central retinal venule equivalent (CRVE) and the central retinal arteriole equivalent (CRAE) dataset[40], within our iROP dataset. Using the summary statistics from the exome chip meta-analysis for CRVE and CRAE, we tested all CRVE/CRAE loci with $p \leq 0.01$ for ROP association within our iROP dataset. A total of 583 SNPs from the CRVE/CRAE dataset were available for analysis in our iROP dataset. After Bonferroni correction for multiple testing, we identified significant associations for SNPs rs13079478 (*FYCO1* gene), rs33910087 (*FYCO1* gene), and rs12357206 (*ANK3* gene) with ROP severity as noted in Table 6.

**Extension of associations for ROP-associated SNPs identified in a candidate gene studies replicate significance of the *RELN* and *EPAS* genes.** To independently replicate candidate-study identified ROP SNPs, we first evaluated candidate SNPs found to be significantly associated with severe ROP by Hartnett et al.[16] within our iROP GWAS dataset. As noted in Table 7, rs10251365 in the gene *RELN* is significantly associated with ROP severity in both cohorts ($p = 0.009$). The $p$-values of SNP associations with ROP severity based on stage using our data are included for comparisons and confirmation. We further sought to validate candidate SNP associations reported within sufficiently powered candidate studies of diverse ethnic and racial backgrounds as noted in Supplementary Table 1. We identified cross-significance for rs13419896 in *EPAS1* ($p = 0.049$) within the iROP cohort for association with ROP disease severity[19].

**GLI3 and other top-associated genes are expressed in human retinal and retinal pigment epithelial tissues (RPE)/Choroid.** To determine the potential relevance of *GLI3* and other top ROP-associated genes to the ocular microenvironment, we sought to determine the expression of these genes within an independent expression dataset generated from human retinal and RPE/choroid tissues as these are the primary tissues affected by ROP disease. Human donor macular or peripheral retinal and RPE/choroid tissues from control eyes ($n = 10$), average age 72 years, within our published eye repository, were analyzed using RNA-sequencing as previously reported by our group[41–44]. Expression of our top-associated ROP SNPs as listed in Table 2 was assessed within both the retinal and RPE/choroid tissues. All top genes except PRPF4B were expressed in human donor macular tissues; the tissue type with greatest expression and fold difference between tissue types (neurosensory retina or RPE/choroid) is listed in Table 8. As noted, GLI3 was expressed in both the neurosensory retina and RPE/choroid; macular RPE/choroid expression was significantly greater by 2.54-fold ($p = 3.3 \times 10^{-15}$, after correction for multiple testing using Benjamini–Hockberg) than neurosensory retinal expression (Table 8). We also analyzed differential GLI3 expression between macular and peripheral RPE/choroid tissues to determine the region with greatest expression. Interestingly, we found that GLI3 expression was 1.72-fold higher in the peripheral RPE/choroid as compared to the macular RPE/choroid ($p < 0.01$) which is notable given the presence of ROP most commonly within the peripheral retina.

## Discussion

Herein, we report a GWAS for ROP which identifies a novel ROP risk association for the *GLI3* SNP rs2058019, reaching genome-wide significance within a multiethnic population. To the best of our knowledge, this is the first reported genome-wide significant SNP association for ROP, which shows an association trend in the same direction for both Hispanic and European Ancestry infants and therefore, may demonstrate disease applicability across populations. We further identify significance of *GLI3* variation for pre-retinal neovascular disease and established ROP genetic risk associations using extension analysis within an independent validation cohort consisting of Hispanic individuals with diabetic retinopathy. We also examined previously reported candidate genes from a severe ROP candidate dataset[16]. While the GWAS data are important alone, we sought to also draw relevance to the disease process. We adopted multiple approaches to validate our findings and demonstrate applicability of our systemic findings to the ocular microenvironment. Using publicly available peripheral retina expression data within the EyeGEx database, we demonstrated possible eQTL functions for several of our top ROP-associated genes. We then also directly evaluated expression of our top-associated genes within RNASeq data in an independent dataset generated by our team from human donor macular retina and RPE tissues. This allows comprehensive analysis within the disease relevant ocular tissues, including both peripheral retinal and macular retinal and retinal pigment epithelial (RPE) expression profiles. Together, these approaches support likely functional and possibly regulatory roles for identified variants and

---

### Table 6 Extension of CRVE/CRAE significant loci to iROP dataset.

| SNP | Gene | iROP | CRVE/CRAE |
|-----|------|------|-----------|
|     |      | **P** | **P** |
| rs13079478 | *FYCO1* | 7.96e-6 | 3.0E-03 |
| rs33910087 | *FYCO1* | 1.93e-5 | 3.0E-03 |
| rs12357206 | *ANK3* | 5.07e-5 | 9.0E-05 |

SNPs (identified for CRVE/CRAE from the literature) confirmed to be significantly associated with ROP Stage.

---

### Table 7 Association of SNPs identified with severe ROP.

| SNP | MAF | Gene | *P* for severe ROP (Harnett, 2014) | *P* for ROP ≥ Stage 3 (iROP) |
|-----|-----|------|------------------------------------|------------------------------|
| rs9332681 | Na | *F5* | 0.99 | Na |
| rs379489 | 0.287 | *CFH* | 3.8E-03 | 0.14 |
| rs395544 | 0.287 | *CFH* | 5.4E-03 | 0.14 |
| rs11587174 | 0.002 | *F13B* | 0.87 | na |
| rs1467199 | 0.211 | *STAT1* | 0.82 | 0.81 |
| rs34417936 | 0.014 | *IL* | 0.66 | Na |
| rs2299386 | 0.380 | *RELN* | 1.6E-03 | 0.73 |
| rs10251365 | 0.332 | *RELN* | 1.0E-03 | 0.0092* |
| rs16879811 | 0.309 | *NRG1* | 4.0E-04 | 0.28 |
| rs16879814 | 0.311 | *NRG1* | 3.0E-04 | 0.25 |
| rs2353512 | 0.002 | *BDNF-AS* | 0.77 | Na |
| rs7127507 | 0.295 | *BDNF* | 2.0E-04 | 0.70 |
| rs2049046 | 0.479 | *BDNF* | 3.00E-05 | 0.41 |
| rs7934165 | 0.471 | *BDNF* | 2.00E-05 | 0.53 |
| rs12281784 | 0.101 | *LRP4* | 0.54 | 0.14 |
| rs9989002 | 0.211 | *IGF1* | 2.2E-03 | 0.81 |
| rs11620315 | 0.149 | *FLT1* | 0.45 | 0.76 |
| rs1319859 | 0.424 | *IGF1R* | 0.12 | 0.04 |
| rs884636 | 0.040 | *PGPEP1L* | 0.10 | na |
| rs7204874 | Na | *NSMCE1jIL4 R* | 1.4E-03 | na |
| rs2057768 | 0.287 | *IL4R* | 7.0E-04 | 0.65 |
| rs1551005 | Na | *TTR* | 0.96 | na |

*MAF* minor allele frequency, *na* SNP not available in ROP dataset and/or with MAF less than 0.05.

---

**Table 8 Top-gene RNA-sequencing-based expression in human donor macular retinal or RPE/choroid tissue isolated using the Utah Protocol.**

| hg19 position | Gene name | Tissue type with higher expression | Fold change between RPE and retina | P-value |
|---|---|---|---|---|
| chr16:89985572-90002500 | TUBB3 | Neurosensory Retina | 35.201 | 1.03E-130 |
| chr8:99037078-99058697 | RPL30 | RPE | 2.974 | 5.69259E-61 |
| chr13:36345477-36705443 | DCLK1 | Neurosensory Retina | 10.303 | 6.27566E-29 |
| chr7:103112230-103629963 | RELN | Neurosensory Retina | 10.599 | 2.41595E-28 |
| chr7:21467651-21554440 | SP4 | Neurosensory Retina | 2.467 | 2.09322E-23 |
| chr16:90014332-90034468 | DEF8 | Neurosensory Retina | 1.709 | 1.10865E-20 |
| chr7:90013034-90142716 | CLDN12 | Neurosensory Retina | 2.050 | 7.47808E-16 |
| chr7:42000547-42277469 | GLI3 | RPE | 2.454 | 3.30197E-15 |
| chr9:8314245-10612723 | PTPRD | Neurosensory Retina | 3.962 | 5.21447E-13 |
| chr2:162848750-162931052 | DPP4 | RPE | 2.725 | 2.96267E-08 |
| chr13:111766158-111768025 | ARHGEF7 | Neurosensory Retina | 2.276 | 3.91738E-05 |

Differential expression between macular retinal and RPE/Choroid was calculated and p-values corrected for multiple testing using Benjamini–Hockberg. The absolute fold change is represented between the tissue with higher expression versus the tissue with lower expression.

corresponding genes within the ocular microenvironment, most relevant to ROP disease. This is particularly true for GLI3 expression which we show is most highly expressed within peripheral tissues, the primary site for ROP pathogenesis.

Owing to the multiethnic composition of our cohort, our analysis also demonstrates potential racial and ethnic differences in GWAS-determined SNP associations. For example, while our lead SNP, rs2058019, demonstrates a less significant association within the African American populations, as noted in Supplementary Table 3, we identify several SNPs with significant association within only the African American population compared with the Hispanic or non-Hispanic European Ancestry populations. This is consistent with, and indeed may speak to, observed clinical differences in ROP incidence and severity, including greater disease development and severity in European Ancestry populations[21,22]. However, while suggestive of disparate genetic risk for ROP disease within different racial and ethnic populations, these results require replication in a larger population. These future analyses will ensure equitable translational value from greater insight into genetic disease risk toward improved patient outcomes for at-risk infants.

The association of GLI3 with ROP disease is novel, though it has a defined role within ocular development. Specifically, the dual role for Gli3 as both a transcriptional activator and repressor of canonical sonic hedgehog (Shh) signaling, has been found to control differentiation of the RPE and rod photoreceptor layer during Xenopus as well as Medaka fish eye development[45,46]. Further, abnormal expression of the Hedgehog signaling pathway has been shown in ROP within the oxygen induced retinopathy (OIR) murine model[47]. More broadly Gli3 has been shown to regulate both the innate and adaptive immune response, with described roles in fetal CD4 and CD8 thymocyte and T-cell development[48,49]. Certainly, pre-retinal neovascular disease, including within ROP and diabetic retinopathy, has been associated with aberrant inflammation[50–52]. Thus, there is a feasible relationship between GLI3 and ROP disease incidence and severity.

The lead SNP, rs2058019, is present within an intronic region and has not previously been associated with disease. Thus, the functional significance is not clear, although aberrancy in GLI3 is associated with human disease, including polydactly syndromes[53] and malignancy[54–57]. Functionally, both aberrant GLI3 repressor and activator roles have been associated with underlying pathomechanisms, resulting in imbalance within epithelial to mesenchymal transition[55,57,58], AKT/ERK1 activation[59], p53 function[60] and autophagy[61]. Significant differences in GLI3

expression between control and diseased tissues is also thought to underlie pathobiology as described in AML[62] and colon cancer[60], in some cases regulated by differential methylation[58,62]. Thus, there is significant precedent for aberrancy of GLI3 DNA structure, expression, or function in human disease. Future work will seek to determine the functional significance of this SNP association to ROP disease development in preterm infants. Our work demonstrating GLI3 expression within adult human donor eyes within both neurosensory retina and RPE/choroid tissues, supports the premise that GLI3 has relevance to human pre-retinal neovascular disease. Additional studies are needed to clarify age-dependent changes in GLI3 expression with relevance to the ROP-affected age range.

In addition to GLI3, genes associated with top-identified SNPs within our dataset, as demonstrated in Table 2, have shown associations with ROP in other work. ARHGEF7, demonstrates significant differential expression within rodent models of pre-retinal neovascularization and is also different between early and late stages of the disease within the OIR model of ROP[63]. There is also precedent for DPP4 function within retinal vascular homeostasis which is perturbed in murine models of both ROP and diabetic retinopathy. Within the OIR model, DPP4-inhibition increased retinal vascularity and leakage while in the murine diabetic retinopathy model, DPP4-inhibition increased retinal vascular leakage[64]. These murine findings are substantiated in human literature, which demonstrates increased progression of proliferative diabetic retinopathy in patients taking DPP4-inhibitors[65]. The inverse was also found to be true, namely that topical administration of dipeptidyl peptidase prevents vascular leakage in a diabetic mouse model[66]. Finally, while CLDN12 has not been found to play a role in pre-retinal neovascular disease or ROP specifically, the Claudin family of proteins have precedent in neovascular ocular disease[67,68]. This is particularly interesting given the identification of another Claudin family member, CLDN14, with a significant SNP association for ROP severity in African Americans (Supplementary Table 3), though this was determined with a significantly reduced number of observations compared to the full cohort and therefore requires replication within a larger diverse cohort.

Genes associated with our top-identified SNPs also support findings from case control candidate-approach SNP-chip studies. While candidate-approach-identified association is often incompletely replicated at the genome-wide level[69–71], as noted in Table 8, extension of associations for the top 10 SNPs identified for severe ROP by Hartnett et al.[16] demonstrates independent replication for rs10251365 in the RELN gene ($p = 0.009$). We also

identified cross-significance for ROP severity with the variant rs13419896 in *EPAS1* ($p = 0.049$)[19] through analysis of previously identified ROP-associated loci from all sufficiently powered candidate gene studies as noted in Supplementary Table 1. While additional cross-significance was not identified, the study design, GWAS versus candidate approach, and outcome measures, "severe compared to mild ROP" versus ROP severity by stage, differ between these studies. As noted above, we also identified the greatest association for these top SNPs within a multiethnic population which is distinct from the roughly 70% Black race population examined by Hartnett et al. Further, we identify unique genetic ROP risk for African American versus European Ancestry or Hispanic populations. As African American unique SNPs were identified in a smaller proportion of our cohort which consisted of fewer observations, these findings warrant examination within larger populations. This is particularly true given the finding that LD was not the same for all racial and ethnic groups as noted in Table 4. Thus, the lack of additional overlap with published candidate-approach-identified loci may also be informed by racial and ethnic differences between studied populations. Alternatively and/or additionally, these differences may reflect other aspects of ROP risk, such as timing of ROP presentation as well as susceptibility of disease progression given external stresses and variability in NICU care practices. Taken together, our identified SNP associations demonstrate relevance to described ROP patho-mechanisms, while also identifying novel associations for further study. Our work further highlights the need for future studies and subsequent meta-analysis to better harmonization and independently validate these findings.

The overall relevance of genetic variation to ROP risk has been debated[72]. Our work substantiates genetic ROP risk which is further highlighted by our genetic risk score assessment. This analysis methodology has been used in numerous disease contexts, including by our group[37,73], to assess the magnitude of genetic disease risk. In our analysis, ROP disease severity was more significantly associated with polygenic SNP subsets than with the most significant individual GWAS SNPs. This suggests the existence of more extensive genetic variation contributing to ROP and thus, the overall importance of this continued approach. However, as our study is the only reported ROP GWAS to date, our analysis represents relative effects rather than absolute estimates of potential genetic association. Additional ROP GWAS studies are needed to ensure that r2 calculations are not overfit and reflect true genetic risk estimates.

Finally, the relevance of our data to vascular pathobiology more broadly is evidenced in our extension analyses within independent replication datasets from other forms of pre-retinal neovascular (DR/GOLDR) and retinal vascular (CVRE/CRAE) pathobiology. We identify SNP associations that bridge these disease contexts and thus may speak to co-occurring mechanisms of disease. Among these, SNP associations at the *GLI3, DCLK1, SP4, PTPRD, RPL30* and *RIDA* genes were replicated in an independent Hispanic diabetic retinopathy (DR) cohort and thus shared within the context of pre-retinal neovascular disease which is present in both ROP ≥ stage 3 and proliferative diabetic retinopathy. We further demonstrate cross-significance of multiple top ROP-associated loci through LD, including for *GLI3*. Thus, while individual SNPs may vary, these data suggest relevance of the top-associated gene regions for pre-retinal neovascular pathobiology. Further, we find existing relevance for several of these regions to retinal pathobiology. As is true for *GLI3*, the *SP4* gene locus has potential relevance to retinal disease given evidence for its role in transcriptional regulation of multiple photoreceptor restricted genes[74,75] Similarly, *DCLK1*[76] and *PTPRD*[77], which demonstrate both SNP and loci-level association within the iROP and GOLDR cohorts, have been previously associated with

DR in animal and human studies respectively. The remaining associations are novel within the context of pre-retinal neovascular disease and represent areas of future study. Similarly, our identification of SNPs within *FYCO1* and *ANK3* as shared between vascular pathobiology in the CVRE/CRAE dataset and our iROP cohort, represents novel associations. Both genes have associations within the eye and retina, though most significantly with anterior segment pathology including infantile cataract and coloboma[78,79].

Taken together, we report an ROP GWAS, which identifies a novel locus at *GLI3* on chromosome 7 demonstrating genome-wide level significance, the first ROP-associated variant to do so to the best of our knowledge. We further identify 9 additional loci with significance of $<5 \times 10^{-6}$ for ROP severity after correction for birth weight and gestational age. We further present multiple lines of evidence demonstrating potential relevance for *GLI3* and other top-associated genes to pre-retinal neovascular pathobiology and the disease microenvironment through *in-silico* extension analyses and expression profiling in human donor eye tissues. Significance varied by race, with greater association measured within European Ancestry and Hispanic compared with African American infants. This may underlie clinically observed differences in ROP development, which are also represented in our cohort which demonstrates African American ROP prevalence of 43% compared to 52% and 67% for European Ancestry and Hispanic populations respectively. Replication of this study using a larger and more diverse population is needed to validate these findings, particularly with regard to differences in genetic ROP risk relative to ethnicity and race[80–82]. In summary and to the best of our knowledge, we report the largest ROP GWAS to date, identifying a novel GWS locus at *GLI3* with possible functional relevance supporting translation toward improved patient outcomes for at-risk preterm infants.

## Methods

**Recruited study cohort**. The Imaging and Informatics for ROP (iROP) study is a multicenter, prospective, ROP cohort study collecting clinical, biologic, and imaging data from male and female preterm infants born with GA and BW risk for ROP as previously published by our group[8,24]. For all enrolled infants, retinal images were obtained using a wide-angle fundus camera [RetCam; Natus Medical Incorporated, Pleasanton, CA]; biologic specimens included blood or saliva and was obtained with specific consent. Informed consent was provided to the parent/guardian under approved IRB protocols at each study site which conformed to the tenets of the Declaration of Helsinki. When consent was given for biologic sample collection blood was collected in a purple top microtainer tube. White blood cells within the buffy coat were isolated using a standard protocol and DNA extracted using a Qiagen Allprep system for subsequent genetic analysis. All images were deidentified in accordance with HIPPA privacy rules. ROP phenotype was determined for each eye by consensus from at least 3 trained graders with ROP expertise employing image-based diagnoses and the clinical exam diagnosis at each study center as previously described[25]. ROP severity was graded based on the worst stage present for either eye. The iROP database was reviewed to identify infants with both biologic and phenotypic data availability which included 920 infants. Patients without detailed demographic, ROP screening, clinical, or imaging data were excluded.

**Genotype, quality control, and imputation**. Genotyping was performed using genotyping arrays from Illumina Infinium Global Screening Array (GSA, Santa Clara, CA, USA). Principal component analysis (PCA) was performed using SMARTPCA

implemented in EIGENSOFT[83]. Standard quality control procedure was performed for all genotype samples/SNPs. For example, samples with genotyping rate less than 0.92, gender mismatches, PCA outliers defined as greater than 3 standard deviations of top principal component variables (PCs), SNPs with genotyping rate less than 0.95 and Hardy-Weinberg Equilibrium (HWE) $p$-value less than $10^{-6}$, were excluded for analysis. Imputation was conducted by Minimac using the Michigan Imputation Server[84] with the reference panel from the 1000 Genomes Project Phase 3 Haplotypes[85]. Autosomal SNPs with the minor allele frequency (MAF) greater than 0.05 and imputation quality (rsq) greater than 0.3 were analyzed.

**GOLDR cohort (Genetics of Latino diabetic retinopathy, GOLDR).** The GOLDR study is a family-based study assessing diabetes and diabetic complications in families (siblings and/or parents) of a proband, defined as having type 2 diabetes and either known DR or a diabetes duration of ≥10 years[31]. Participants are all Latinos of Mexican or Central American origin, recruited, and studied between 2007 and 2012 at the Lundquist Institute (formerly called the Los Angeles BioMedical Research Institute) at Harbor-UCLA Medical Center (HUMC). In total, there were 612 participants with type 2 diabetes from 216 families, with sizes ranging from 1 to 8 members per family in the study. These samples were genotyped with the Illumina Cardio-Metabochip and imputed with 1000 Genome using Michigan Imputation Server.

**Genome-wide association tests for single SNPs.** We employed standard GWAS methodology which have been used successfully for datasets of this sample size. Under the additive genetic model, we performed GWAS tests with allele dosage for all genotyped and imputed SNPs stratified by each ethnic group, and also combined, using Efficient Mixed Model Association eXpedited (EMMAX) method[86] implemented in EPACTS (https://genome.sph.umich.edu/wiki/EPACTS). Outcome measure for association was ROP severity defined by stage 3 or higher; the overall approach was case control with cases defined as infants with ROP stage 3 or greater and Controls defined as infants with Stage 2 or less severe disease. Birth weight, gestational age, gender, top 3 PCs, and the relatedness by the genomic relationship matrix estimated using the genomic data, were included as covariates. Associations with $p < 5 \times 10^{-8}$ were considered GWAS significant. To further investigate the relationship between birth weight/gestational age and ROP association signals, we also performed the association tests with and without adjustment for birth weight/gestational age. In addition, GWAS analyses stratified by each ethnic or racial group were also performed in the same way.

**Retinal expression quantitative trait loci (eQTL) analysis.** To investigate the relationship between association signals and gene expression, we evaluated whether top ROP SNPs identified from GWAS are related to eQTL in the genotype-tissue specific expression databases, EyeGEx[30], by far the largest reference tissue bank for gene expression and regulation in the human retina.

**Genetic risk score (GRS) of ROP.** Three genetic risk scores were tested. For the "GLI3" risk score was composed of the genotype for the most significant SNP, rs2058019. In order to reduce overfitting of the genetic risk score to ROP stage due to the large number of SNPs relative to the number of subjects, we fit a linear regression model with a LASSO penalty using the R package "glmnet" with 451,565 directly genotyped SNPs[87,88]. The outcome variable was ROP Stage as an ordinal categorical variable (None, Stage 1, Stage 2, and Stage 3 and above). The glmnet

software performs 10-fold cross-validation of the mean-squared error with a series of 100 penalties selected by a coordinate-descent; two models result, SNP score B was composed of 177 SNPs with MAF 0.05 selected at the minimum penalty, and SNP score A was composed of 33 SNPs with MAF 0.05 at 1 standard error (1 s.e.) from the mean-squared error. Selecting SNPs at 1 s.e. is an additional way to reduce overfitting. Using the weights from our GWAS study above, we then calculated the weighted GRSs for SNP score B and SNP score A, defined as the weighted sum of the number of risk alleles across all the SNPs for that score. Scores were tested in a linear model with sex, birth weight, gestation age, and the first two principal components of the SNP data. Principal components were calculated using a singular value decomposition. As there is no other reported ROP GWAS, our weighted analysis was performed within the same iROP dataset therefore overfitting $r^2$.

**Extension of the association signals for diabetic retinopathy in GOLDR.** Since many infants are Hispanic origin, we further performed the SNP association tests in an independent Hispanic cohort (GOLDR). Using diabetic retinopathy (DR) patients as cases (72 cases vs 491 controls), the association tests were performed for the top ROP association genes/loci identified above (10 loci) in GOLDR cohort with the logistic regression. The significance threshold for the extension analysis was defined by Bonferroni correction with the number of genes/loci tested (0.05/10 = 0.005).

**Extension of association signals for CRVE/CRAE loci.** Further extension analysis was performed to cross-check the overlapping SNP associations with CRVE and/or CRAE. Using the summary statistics from the exome chip meta-analysis for CRVE and CRAE[40], we evaluated association signals for all available CRVE/CRAE loci with $p$-values < 0.01 (583 SNPs in total) in ROP dataset. The significance threshold for the extension analysis was defined by Bonferroni correction with the number of SNPs tested (0.05/583 = $9 \times 10^{-5}$).

**Extension of the association signals for severe ROP identified by Hartnett et al. candidate gene study.** In order to replicate the ROP association signals by other studies, we tested the significant SNPs identified by Hartnett et al.[16] associated with severe ROP. Out of 22 top severe ROP signals, we evaluated 15 SNP associations available in our analysis. The significance threshold for the extension analysis was defined by Bonferroni correction with the number of SNPs tested ($p = 0.05/15 = 0.0033$).

**Donor eye tissue repository.** Methods for human donor eye collection were previously described in detail according to a standardized protocol[41]. In brief, in collaboration with the Utah Lions Eye Bank, donor eyes were procured within a 6-h post-mortem interval, defined as death-to-preservation time. Both eyes of the donor underwent post-mortem phenotyping with ocular imaging, including spectral domain optical coherence tomography (SD-OCT), and color fundus photography as published. Retinal pigment epithelium/choroid was immediately dissected from the overlying retina, and macula separated from periphery using an 8 mm macular punch. For both peripheral and macular tissues, RPE/choroid was separated from the overlying retinal tissue using microdissection; tissue planes were optimized to minimize retinal contamination of RPE/choroid samples using a subsequent 6 mm RPE/choroid tissue punch. Isolated macular and peripheral RPE/choroid samples were preserved in RNAlater (Ambion, ThermoFisher, Waltham, MA, USA), and stored at -20 °C for 24 h then transferred to -80 °C. Phenotype analysis was

performed as described[41] by a team of 4 retinal specialists and ophthalmologists at the University of Utah School of Medicine, Moran Eye Center and the Massachusetts Eye and Ear Infirmary Retina Service. Agreement of all 4 specialists upon independent review of the color fundus and OCT imaging was deemed diagnostic; discrepancies were resolved by collaboration between a minimum of three specialists to ensure a robust and rigorous phenotypic analysis. One eye was per donor was biochemically analyzed. Institutional approval, and the consent of patients to donate their eyes and for research purposes was obtained from the University of Utah and conformed to the tenets of the Declaration of Helsinki. All tissue was deidentified in accordance with HIPPA privacy rules.

**Nucleic acid extraction and RNA-Sequencing.** Transcriptional profiling of macular and peripheral retina and RPE/choroid tissues from 10 unrelated control eye tissue donors, average age 72 years, was performed using RNA-sequencing. DNA and RNA were extracted from peripheral and macular neurosensory retinal and RPE/choroid tissues, prepared as described above, using the Qiagen Allprep DNA/RNA mini kit (cat #80204) per the manufacturer's protocol. Quality of RNA samples was assessed with an RNA Nano Chip (Agilent). Total RNA was poly-A selected and cDNA libraries were constructed using the Illumina TruSeq Stranded mRNA Sample Preparation Kit (cat# RS-122-2101, RS-122-2102) according to the manufacturer's protocol. Sequencing libraries (18 pM) were chemically denatured and applied to an Illumina TruSeq v3 single read flowcell using an Illumina cBot. Hybridized molecules were clonally amplified and annealed to sequencing primers with reagents from an Illumina TruSeq SR Cluster Kit v3-cBot-HS (GD-401-3001). Following transfer of the flowcell to an Illumina HiSeq instrument (HCS v2.0.12 and RTA v1.17.21.3), a 50-cycle single read sequence run was performed using TruSeq SBS v3 sequencing reagents (FC-401-3002).

**Primary analysis of RNA-sequencing data.** Each of the 50 bp, poly-A selected, non-stranded, Illumina HiSeq fastq datasets were processed as follows: Reads were aligned using NovoCraft's novoalign 2.08.03 software (http://www.novocraft.com/) with default settings plus the -o SAM -r All 50 options to output multiple repeat matches. The genome index contained human hg19 chromosomes, phiX (an internal control), and all known and theoretical splice junctions based on Ensembl transcript annotations. Additional details for this aspect of the protocol are described elsewhere (http://useq.sourceforge.net/usageRNASeq.html). Next, raw novoalignments were processed using the open source USeq SamTranscriptiomeParser (http://useq.sourceforge.net) to remove alignments with an alignment score greater than 90 (~3 mismatches), convert splice junction coordinates to genomic, and randomly select one alignment to represent reads that map equally well to multiple locations. Relative read coverage tracks were generated using the USeq Sam2USeq utility (http://useq.sourceforge.net/cmdLnMenus.html#Sam2USeq) for each sample and sample type (Normal Retina, Neovascular AMD Retina, Intermediate AMD Retina, Normal RPE, Neovascular AMD RPE, and Intermediate AMD RPE). These data tracks are directly comparable in genome browsers and good tools to visualize differential expression and splicing. Estimates of sample quality were determined by running the Picard CollectRnaSeqMetrics application (http://broadinstitute.github.io/picard/) on each sample. These QC metrics were then merged into one spreadsheet to identify potential outliers. Agilent Bioanalyzer RIN and library input concentration columns were similarly added for QC purposes (http://www.genomics.agilent.com).

**Statistics and reproducibility.** Genetic analyses, ie GWAS associations, were performed mainly in the Linux environment. All p-values presented are based on two-sided tests. Software includes EPACTS, SMARTPCA, PLINK, and GLMNET. GWAS significance threshold is $p \leq 5 \times 10^{-8}$ accounting for multiple testing with millions of SNPs. The significance threshold of replication associations is using the Bonferroni approach adjusting for the number of independent tests. Statistical analyses of gene expression within human donor tissues were conducted using paired t-testing and p-values corrected for multiple testing using Benjamini–Hockberg.

**Reporting summary.** Further information on research design is available in the Nature Portfolio Reporting Summary linked to this article.

## Data availability

GWAS summary statistic data are publicly available in dbGAP; accession number phs002020.v1.p1. Processed RNA-sequencing data presented herein are publicly available in ZenodoData: https://doi.org/10.5281/zenodo.7532115, and ZenodoData: https://doi.org/10.5281/zenodo.10161540. The raw RNA-Sequencing data reported in this study cannot be deposited in a public repository due to ethical concerns. To request access, email Margaret M. DeAngelis (mmdeange@buffalo.edu) or Leah Owen (leah.owen@hsc.utah.edu).

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

## Acknowledgements

Supported by the National Institutes of Health, National Eye Institute (NIH/NEI) Contract RO1EY019474, R01 EY031331, R21 EY031883, and P30 EY10572 the National Center for Advancing Translational Sciences, CTSI grant UL1TR001881, and the National Institute of Diabetes and Digestive and Kidney Disease Diabetes Research Center (DRC) grant DK063491 to the Southern California Diabetes Endocrinology Research Center. unrestricted departmental funding and a Career Development Award (Dr Campbell) from Research to Prevent Blindness (New York, NY), and with support from the US Agency for International Development, the Seva Foundation, and Helen Keller International. Infrastructure for the CHARGE Consortium is supported in part by the National Heart, Lung, and Blood Institute (NHLBI) grant R01HL105756. L.A.O.: 1K08EY031800-03; (PI). This work was supported by National Institutes of Health Core Grant (EY014800), and an Unrestricted Grant from Research to Prevent Blindness, New York, NY, to the Department of Ophthalmology & Visual Sciences, University of Utah. M.E.H.: R01EY015130 and R01EY107011 (PI). R21: R21EY033579 (co-PI). M.M.D.: Ira G. Ross and Elizabeth Olmsted Ross Endowed Chair. M.M.D. receives research grant support from Genentech (San Francisco, CA). This paper was supported in part by the Genetics of Latinos Diabetic Retinopathy (GOLDR) Study grant EY14684. This work is supported in part by the Intramural Research Program of the NIH (National Eye Institute).

## Author contributions

Conceptualization: J.I.R., X.L., K.D.T., E.I., M.F.C., R.V.P.C., J.P.C., S.O., L.A.O., M.M.D. Methodology: J.I.R., X.L., K.D.T., Y.D.I.C., X.G., E.I., M.M.D., J.P.C. Formal Analysis: X.L., K.D.T., X.G., M.M.D., L.A.O. Investigation: S.O., J.P.C., M.F.C., R.V.P.C., J.I.R., E.I., K.R., P.G. Resources: J.I.R., E.I., K.R., P.G., Y.D.I.C., M.M.D., L.A.O., J.P.C., M.F.C., R.V.P.C. Data Curation: X.L., X.G., K.D.T. Writing – Original Draft Preparation: L.A.O., J.I.R., X.L., J.P.C. Critical Revision: J.I.R., X.L., K.D.T., E.I., M.F.C., J.P.C., S.O., L.A.O., M.M.D., R.V.P.C., M.E.H., Y.D.I.C., X.G., A.C., K.S. Supervision: J.P.C., J.I.R., Y.D.I.C., E.I., X.G., M.M.D. Project Administration: J.I.R., J.P.C. Funding Acquisition: J.I.R., J.P.C., M.F.C., R.V.P.C., M.M.D.. iROP consortium members who did not directly contribute to this work though have contributed to the overall consortium work are listed here: B.K.Y., S.J.K., R.S., K.J., B.K., J.H., O.C., C.E., L.S., A.O., A.B., C.N., K.D., K.C., T.O., T.C., M.Z., T.L., A.N., E.K., K.M., D.C., M.H., C.S., R.M., S.G., L.L., D.M., M.N., Z.W., J.K., D.E., S.I., M.A.M., S.S.L., R.R., A.A., F.O.M., M.M.G., C.M.D., C.M.M.

## Competing interests

J.P.C. receives research support from Genentech (San Francisco, CA). J.P.C. was a consultant to Boston AI Lab (Boston, MA). R.V.P.C. is on the Scientific Advisory Board for Phoenix Technology Group (Pleasanton, CA), a consultant for Alcon (Ft Worth, TX). M.F.C. was previously a consultant for Novartis (Basel, Switzerland), and was previously an equity owner of InTeleretina, LLC (Honolulu, HI). J.P.C. and R.V.P.C. are equity owners of Siloam Vision. M.M.D. receives research grant support from Genentech (San Francisco, CA). The remaining authors declare no competing interests.

## Additional information

## on behalf of the i-ROP Consortium

J. Peter Campbell[15], Susan Ostmo[15], Aaron Coyner[15], Benjamin K. Young[15], Sang Jin Kim[15], Kemal Sonmez[15], Robert Schelonka[15], Michael F. Chiang[13], R. V. Paul Chan[16], Karyn Jonas[16], Bhavana Kolli[16], Jason Horowitz[17], Osode Coki[17], Cheryl-Ann Eccles[17], Leora Sarna[17], Anton Orlin[18], Audina Berrocal[19], Catherin Negron[19],

Kimberly Denser[20], Kristi Cumming[20], Tammy Osentoski[20], Tammy Check[20], Mary Zajechowski[20], Thomas Lee[21], Aaron Nagiel[21], Evan Kruger[21], Kathryn McGovern[21], Dilshad Contractor[21], Margaret Havunjian[21], Charles Simmons[22], Raghu Murthy[22], Sharon Galvis[22], Jerome Rotter[23], Ida Chen[23], Xiaohui Li[23], Kent Taylor[23], Kaye Roll[23], Leah Owen[24], Lucia Lucci[24], Mary Elizabeth Hartnett[25], Darius Moshfeghi[25], Mariana Nunez[25], Zac Weinberg-Smith[25], Jayashree Kalpathy-Cramer[26], Deniz Erdogmus[27], Stratis Ioannidis[27], Maria Ana Martinez-Castellanos[28], Samantha SalinasLongoria[28], Rafael Romero[28], Andrea Arriola[28], Francisco Olguin-Manriquez[28], Miroslava Meraz-Gutierrez[28], Carlos M. Dulanto-Reinoso[28] & Cristina Montero-Mendoza[28]

[15]Oregon Health & Science University, Portland, OR, USA. [16]University of Illinois at Chicago, Chicago, IL, USA. [17]Columbia University New York, New York, NY, USA. [18]Weill Cornell Medical College New York, New York, NY, USA. [19]Bascom Palmer Eye Institute, Miami, FL, USA. [20]William Beaumont Hospital Royal Oak, Royal Oak, MI, USA. [21]Children's Hospital Los Angeles Los Angeles, Los Angeles, CA, USA. [22]Cedars Sinai Hospital Los Angeles, Los Angeles, CA, USA. [23]Lundquist Institute Torrance, Torrance, CA, USA. [24]University of Utah Salt Lake City, Utah Salt Lake City, UT, USA. [25]Stanford University Palo Alto, Palo Alto, CA, USA. [26]Massachusetts General Hospital Boston, Boston, MA, USA. [27]Northeastern University Boston, Boston, MA, USA. [28]Asociacion para Evitar la Ceguera en Mexico (APEC), Mexico City, USA.

