## [Peer Review File · Communications Biology]

Reviewers' comments:

Reviewer #1 (Remarks to the Author):

Li et al. have conducted the largest ROP GWAS to date and found a novel locus at GLI3 associated with ROP. Integrated genetic risk score analysis and expression profiling in eye tissues have revealed GLI3 underlying genetics risk for ROP.

In general, the research was of high scientific quality and the results provided a significant advance in the field. I have some comments below:

- #1. A comprehensive GWAS was conducted for the ROP severity. Have the authors explored the significant variants by some computational algorithms for robustness check, such as fastGWA (Jiang et al., Nature Genetics), BOLT-LMM (Loh et al., Nature Genetics).
- #2. The p-value for almost eQTL association in Table2 were not significant, which did not support the conclusion of "relevance to the ocular microenvironment" mentioned in the paper (Page 6).
- #3. The Supplementary Figure 2 were not included in the material available for review. This prevented a thorough evaluation of the submission.
- #4. I think that independent bulk RNAseq as well as single-cell RNAseq datasets are needed to further validate and investigate the relevance of GLI3 and other top associated genes to retinal pathophysiology.

Reviewer #2 (Remarks to the Author):

Li et al. performed a GWAS in a multiethnic cohort of 920 at-risk infants for retinopathy of prematurity (ROP) based in the i-ROP consortium, which was the largest GWAS for ROP to date. ROP is a retinal vascular disease affecting premature infants and a leading cause of childhood blindness worldwide. From the GWAS, authors identified two loci at the genome-wide significance level and seven loci at the suggestive level. The most significant signal falls in the locus of GLI3 gene, which has been known for its role in the eye development. Following the GWAS, authors examined the co-association of significant SNPs in diabetic retinopathy and the expression of candidate genes in the human retinal and RPE tissues, in order to substantiate the association signals. In addition, authors examined the ethnic difference of SNP associations with ROP severity from the multiethnic cohort of i-ROP consortium including Hispanic, Caucasian, and African-descendants.

The manuscript was well written and the study design was well organized. However, I found several drawbacks needed to be fixed throughout the manuscript as follows.

1. It is not clear for the association study design to identify the genetic variants related to the development of ROP or ROP severity. Did authors apply the case-control study or the categorical study (using the stage number of ROP) for the ROP GWAS? If authors used the case-control, please specify which infants were used for the matching control and also which stages of ROP were considered as cases.
2. It is confusing whether this GWAS identified one genome-wide significant locus or two genome-wide significant loci. While authors mentioned in the abstract that they identified two genome-wide significant loci, Figure 1 of Manhattan plot for the GWAS showed only one genome-wide significant locus that presumably fell in the GLI3 gene.
3. In page 7 of the Results, authors titled one section as "Multiple significant SNPs are identified regionally related to GLI3", which may mislead readers as authors identified several independent signals in the GLI3 locus. I do not see multiple significant signals in Figure 2 of the regional plot but only one independent signal.

4. In page 8 of the Results, authors mentioned Supplemental Figure 2 as the depicted GWASs separately for each ethnic group. However, there is no Supplemental Figure 2 in the Supplemental data file and Supplemental Figure 1 (A) showed the Manhattan plots of the GWASs. It needs to be fixed. Moreover, it is hard to discriminate each signal from the three GWAS results. It may be better to display them separately.
5. I concern the GWAS results derived from association analysis using only the African American (AA) ROP patients, since there might be only 14 cases included for the AA ROP GWAS, estimated from Table 1 by calculating the stage 3 or greater as AA case. Further, the association signals of AA might differ than those of Caucasian (CA) and Hispanic (HA), based on Table 3 and 4 as well as Supplementary Table 3.
6. In Table 5, authors showed the cross-significance of top-associated ROP SNPs with diabetic retinopathy (DR) or vice versa. Since there are many SNPs not available in either GWAS data, the comparisons are not easy to follow. It would be better to provide the linkage disequilibrium (LD) information between ROP top SNPs and DR top SNPs shown in the Table. For example, authors found two SNPs of rs74527981 and rs17172024 close to the GLI3 locus significant in DR GWAS but these SNPs were not available in ROP GWAS data. Are there LD relationship between them and rs2058019, the most significant SNP in the ROP GWAS?
7. In page 9 of the Results, authors investigated the ROP GRS and the trait variance explained by associated SNPs as well as GRSs. The R-squared values in Table 6 seem be larger than those normally found in similar studies. Moreover, the evaluation of GRS should be performed in the independent samples from the GWAS samples. Please provide the detail method for the construction and evaluation of GRS.

Reviewer #3 (Remarks to the Author):

Reviewer Critique

In a study, entitled "Genome-wide association identifies novel ROP risk loci in a multi-ethnic cohort", Dr. Li and colleagues conducted a genome-wide association study (GWAS) in a multiethnic cohort of 920 at-risk infants for retinopathy of prematurity (ROP), reporting two loci at genome wide significance level and seven loci at suggestive significance for ROP stage 3 or higher. The most significant locus identified the Glioma-associated oncogene family zinc finger 3 (GLI3) gene, with the most significant SNP (rs2058019) driven by the Hispanic and Caucasian infants falling in an intronic region of the gene. Relevance for GLI3 and to some extent, the other associated genes, to human ocular disease was further substantiated through in-silico analyses, genetic risk score and expression profiling in human donor eye tissues. The authors conclude that they have conducted the largest ROP GWAS to date, identifying a novel locus at GLI3 with relevance to retinal biology supporting genetic susceptibilities for ROP risk with possible variability by race and ethnicity. While these results are of potential interest to the readership on Communications Biology the authors discovery of these new ROP loci/genes, requires further replication and functional studies before they can be confirmed. I have outlined my other concerns below:

Major Comments:

1. It would be beneficial to the readership if the authors could supply more clinical description of the cases and how ROP diagnosis was made in the context of clinical variables such as supplemental oxygen needs/use, birth weight and GSA, other systemic illnesses, surfactant, corticosteroid and other medication use, and variation in treatment protocols between the 14 centers for these variables to make it explicit how these variables were corrected for vs used as covariates in the analysis.
2. The authors should include confidence intervals (CI) for their odds ratios in Table 2 to make it easier for the reader to determine the significance of the findings reported.

3. In addition to lending support to new GLI3 locus and other loci of suggestive significance, the authors should address how many of the previously reported candidate risk loci were replicated (this is done separately for the RELN locus) – this should be further discussed why the authors think some of these loci did not replicate – if phenotypic differences, lack of power, spurious results previously or other reasons – as this is the best powered study to this date and no previous GWAS locus has reached GWS, this will help clear the state of the existing literature on genetic underpinnings of ROP.

4. The authors state that ...as noted in Table 2, we identified 10 loci which demonstrate genome-wide significance ($p \leq 5 \times 10^{-8}$) or suggestive significance ($p \leq 5 \times 10^{-6}$) of association with ROP severity, defined by stage 3 disease or greater. It should be stated instead that the authors identified one GWS locus and 9 loci of suggestive significance.

5. Replication of these study results in independent cohorts should be conducted in future studies.

Minor comments:

1. The authors should consider imputation with the TopMED dataset in future analyses to more accurately address rare variants and empower the study results.

November 5th, 2023

RE: *Genome-wide association identifies novel ROP risk loci in a multi-ethnic cohort*; CoMMSBIO-23-1548A

Dear Reviewers,

Thank you for your time and insightful comments; incorporation of these critiques has enhanced our work. We have included a detailed point by point discussion of these revisions in our following point by point rebuttal. These changes are reflected in our manuscript as tracked changes.

Sincerely,

Leah A. Owen, MD, PhD

Reviewer #1:

#1. A comprehensive GWAS was conducted for the ROP severity. Have the authors explored the significant variants by some computational algorithms for robustness check, such as fastGWA (Jiang et al., Nature Genetics), BOLT-LMM (Loh et al., Nature Genetics).

Author response: Thank you for this comment, we appreciate the opportunity to clarify our approach. We now include language in our methodology section indicating that for this first of its kind ROP GWAS, we used standardized methodology (lines 482-483). We specifically performed GWAS analysis adjusting for the genetic relationship matrix and principal components to ensure a robust analysis and reduce spurious associations. We also analyzed separately for each ethnic group to ensure this was not confounding. We did not employ additional methods including fastGWA or BOLT-LMM analysis as these were developed for larger sample sizes and may over-correct within our cohort. Given these limitations of our sample size, we adopted a validation approach using transcriptomic data and extension to additional pre-retinal neovascular disease datasets.

#2. The p-value for almost eQTL association in Table 2 were not significant, which did not support the conclusion of "relevance to the ocular microenvironment" mentioned in the paper (Page 6).

Author response: We apologize that this approach and interpretation was not clear in our original submission. For our eQTL analysis, we adopted an understanding that orthogonal data, for example transcriptomic data, can have disease relevance even when not meeting Bonferroni correction. We also used multiple lines of evidence to support relevance of our GWAS findings to the ocular microenvironment. In our revised work we improve our discussion of these points and include the following within the discussion (lines 293-304): *While the GWAS data are important alone, we sought to also draw relevance to the disease process. We adopted multiple approaches to validate our findings and demonstrate applicability of our systemic findings to the ocular microenvironment. Using publicly available peripheral retina expression data within the Eyetex database, we demonstrated possible eQTL functions for several of our top ROP-associated genes. We then also directly evaluated expression of our top-associated genes within RNASeq data in an independent dataset generated by our team from human donor macular retina and RPE tissues. This allows comprehensive analysis within the disease relevant ocular tissues, including both peripheral retinal and macular retinal and retinal pigment epithelial (RPE) expression profiles. Together, these approaches support likely functional and possibly regulatory roles for identified variants and corresponding genes within the ocular microenvironment, most relevant to ROP disease. This is particularly true for GLI3 expression which we show is most highly expressed within peripheral tissues, the primary site for ROP pathogenesis.*

#3. The Supplementary Figure 2 were not included in the material available for review. This prevented a thorough evaluation of the submission.

Author response: The reviewer is correct and this was a mis-labeling on our part. We apologize for this oversight. In our revised submission, the text refers to the correct Supplementary Figure 1.

#4. I think that independent bulk RNAseq as well as single-cell RNAseq datasets are needed to further validate and investigate the relevance of GLI3 and other top associated genes to retinal pathophysiology.

Author response: We appreciate this comment and agree with the reviewer that validation within expression datasets is valuable. We have now better clarified that this is included in Table 9 (now renumbered as Table 8) in our revised manuscript (lines 264-280) and that our evaluation of top ROP-associated gene expression was done in an independent bulk RNAseq dataset generated from macular retinal and RPE/choroid tissue from human primary tissue in our laboratory. These data are now published within the setting of the primary analysis, and we have added the appropriate citation. Please also refer to our comments for critique # 2.

Reviewer #2:

1. It is not clear for the association study design to identify the genetic variants related to the development of ROP or ROP severity. Did authors apply the case-control study or the categorical study (using the stage number of ROP) for the ROP GWAS? If authors used the case-control, please specify which infants were used for the matching control and also which stages of ROP were considered as cases.

Author response: We appreciate the opportunity to clarify these important points. We have added clarifying language in our Results section (lines 112-114) to indicate that our primary GWAS analysis was done using a case control approach designed to determine variation associated with severe ROP. Cases included infants with stage 3 ROP disease or greater and Control infants included those with stage 2 or less severe disease. In addition, as noted in our Methods section, our genetic risk score (GRS) analysis was performed using an ordinal, categorical, approach. For greater clarity, we have also included this information within the Results section, lines 222-225.

2. It is confusing whether this GWAS identified one genome-wide significant locus or two genome-wide significant loci. While authors mentioned in the abstract that they identified two genome-wide significant loci, Figure 1 of Manhattan plot for the GWAS showed only one genome-wide significant locus that presumably fell in the GLI3 gene.

Author response: We apologize for this confusion and have clarified our language within both the abstract and manuscript text (lines 38-39; 126-129) to indicate that although we initially identified 2 SNPs with significance of $p < 5 \times 10^{-9}$, only one SNP, our lead GLI3 SNP, remained significant at this level after correction for the potential confounding effects of birthweight and gestational age.

3. In page 7 of the Results, authors titled one section as "Multiple significant SNPs are identified regionally related to GLI3", which may mislead readers as authors identified several independent signals in the GLI3 locus. I do not see multiple significant signals in Figure 2 of the regional plot but only one independent signal.

Author response: We agree with the reviewer that our caption wording should be revised to more precisely indicate our findings. As such, we have revised this heading to read: **Multiple SNPs are identified regionally related to GLI3.**

4. In page 8 of the Results, authors mentioned Supplemental Figure 2 as the depicted GWASs separately for each ethnic group. However, there is no Supplemental Figure 2 in the Supplemental data file and Supplemental Figure 1 (A) showed the Manhattan plots of the GWASs. It needs to be fixed. Moreover, it is hard to discriminate each signal from the three GWAS results. It may be better to display them separately.

Author response: We apologize for this oversight; in our original submission the text incorrectly referred to Supplemental Figure 1 as Supplemental Figure 2. Our revised submission now correctly cites Supplemental Figure 1. We have additionally revised the Manhattan plot as suggested by the Reviewer such that analysis for each race/ethnicity is displayed separately.

5. I concern the GWAS results derived from association analysis using only the African American (AA) ROP patients, since there might be only 14 cases included for the AA ROP GWAS, estimated from Table 1 by calculating the stage 3 or greater as AA case. Further, the association signals of AA might differ than those of Caucasian (CA) and Hispanic (HA), based on Table 3 and 4 as well as Supplementary Table 3.

Author response: We agree with Reviewer 2 that our analysis within the AA population was limited by a small sample size, particularly with respect to cases. While this is a significant limitation in our ability to draw meaningful conclusions, we also found interesting that the significant SNPs within Hispanic and European Ancestry populations were similar to one another but dissimilar to those within the AA population. While we cannot state this as a conclusion of our study given concerns for power, we sought to use these data to highlight the need for future work to investigate associations relative to race and ethnicity with sufficient power. This understanding will best allow for precision approaches toward efficacious ROP prevention and treatment for all infants. Therefore, to address this point while

appropriately acknowledging the limitations of this analysis, we now include these data as Supplementary Table 4 rather than a primary table and have included these data within the larger Results section rather than as a specific focus (lines 143-162). We also expanded our discussion of these limitations within our Discussion section (lines 358-360; 373-388).

6. In Table 5, authors showed the cross-significance of top-associated ROP SNPs with diabetic retinopathy (DR) or vice versa. Since there are many SNPs not available in either GWAS data, the comparisons are not easy to follow. It would be better to provide the linkage disequilibrium (LD) information between ROP top SNPs and DR top SNPs shown in the Table. For example, authors found two SNPs of rs74527981 and rs17172024 close to the GLI3 locus significant in DR GWAS but these SNPs were not available in ROP GWAS data. Are there LD relationship between them and rs2058019, the most significant SNP in the ROP GWAS?

Author response: We thank the Reviewer for this suggestion as addition of this analysis has strengthened our revised work. As now included in revised Table 5 (now renumbered as Table 4), our results indicate that the top gene loci demonstrate relevance for both conditions characterized by preretinal neovascularization (Diabetic Retinopathy and Retinopathy of Prematurity), although the individual SNPs differ in some cases. As both the GOLDR and iROP cohorts have large Hispanic populations, LD analysis was performed within the Hispanic patients, making the analysis more credible and rigorous. Through incorporation of this analysis, we additionally found the MAF differed within the European Ancestry and African American populations, therefore LD was not the same for all populations. Taken together, this valuable additional suggested analysis 1) increases the strength of association for multiple top-identified loci with the common underlying pathophysiology, preretinal neovascularization and 2) further substantiates the importance of future work investigating potential differences by race and ethnicity toward improved patient outcomes within all ROP at-risk populations. We have included these data within our revised Table 4 and Results section (lines 195-215) as well as within the revised Discussion section (lines 373-388; 408-415).

7. In page 9 of the Results, authors investigated the ROP GRS and the trait variance explained by associated SNPs as well as GRSs. The R-squared values in Table 6 seem be larger than those normally found in similar studies. Moreover, the evaluation of GRS should be performed in the independent samples from the GWAS samples. Please provide the detail method for the construction and evaluation of GRS.

Author response: We agree with the Reviewers comments and have addressed both in our revised Methods and Results sections. Our revised Methods section now more clearly states that the R-squared values in Tables 6 (now re-numbered as Table 5) are not for GRSs only, but after adjusting for important clinical covariates and principal components. In addition, we agree with reviewer's comment that the ideal way of GRS evaluation is using independent samples. However, as there is no other large scale GWAS studies for ROP, we calculated the scores from a group of potential SNPs with a penalized regression model with the 10-fold cross-validation using LASSO method implemented in 'glmnet' package to maximize the likelihood of SNPs' selection contributing to the genetic risk. The purpose of Table 5 is to compare the performance of different GRSs constructed from potential SNP groups in our dataset to the GWS GLI3 SNP, not trait variance explained testing or genetic risk validation in independent samples. In the revised manuscript, we 1) refined the SNP sets selection after removing SNPs with $MAF < 0.05$; 2) calculated weighted GRSs based on weights from our GWAS; 3) noted the r-squared values are inflated without independent samples validation. In this way we clearly denote they indicate relative effects rather than absolute estimates. We have modified the Methods with detailed information (lines 501-515) and now present results with limitation and caution (Results: lines 218-239; Discussion: lines 391-400).

Reviewer #3:

Major Comments:

1. It would be beneficial to the readership if the authors could supply more clinical description of the cases and how ROP diagnosis was made in the context of clinical variables such as supplemental oxygen needs/use, birth weight and GSA, other systemic illnesses, surfactant, corticosteroid and other medication use, and variation in treatment protocols between the 14 centers for these variables to make it explicit how these variables were corrected for vs used as covariates in the analysis.

Author response: We thank the reviewer for the opportunity to clarify these points. In our revised manuscript, we include greater detail regarding phenotyping protocols used in our consortium as well as analysis controlling for the potentially confounding effects of established ROP risk factors including gestational age and birth weight. Specifically, we include additional literature references for our consortium phenotyping protocols, which demonstrate the highest level of rigor within the existing literature. These changes are reflected in our revised Results section (lines 102-104) to augment the details contained within our Methods section. Further, while the NICU practice was not uniform relative the other factors cited by Reviewer 3, we measure the GLI3 SNP association despite this variability which mirrors heterogeneity seen clinically. Thus, this may strengthen confidence in the measured association. Further, our ability to measure significance at the genome wide level with relatively small number of observations comparing to other GWAS studies, supports the stringency of our phenotyping protocols (Klein et al. 2005; Dewan et al. 2006). Certainly, the point is well made, and future work will more precisely correlate our genetic findings with ROP risk within individual NICU populations, ethnicities and races and determine functional significance. These points and potential limitations are now included in our Discussion section, lines 362-388.

2. The authors should include confidence intervals (CI) for their odds ratios in Table 2 to make it easier for the reader to determine the significance of the findings reported.

Author response: We agree with the Reviewer that addition of CI in Table 2 strengthens our manuscript and appreciate the opportunity to include this in our revised work.

3. In addition to lending support to new GLI3 locus and other loci of suggestive significance, the authors should address how many of the previously reported candidate risk loci were replicated (this is done separately for the RELN locus) – this should be further discussed why the authors think some of these loci did not replicate – if phenotypic differences, lack of power, spurious results previously or other reasons – as this is the best powered study to this date and no previous GWAS locus has reached GWS, this will help clear the state of the existing literature on genetic underpinnings of ROP.

Author response: We agree that including additional analysis demonstrating replication of previously reported candidate risk loci in our GWAS cohort strengthens our work. While there has not previously been a reported GWAS for ROP, significantly powered candidate and whole exome approaches have been reported as summarized in Supplemental Table 1. In our revised manuscript, we include additional analysis examining these variants in our dataset which demonstrates cross-significance of candidate identified rs13419896 in *EPAS1* ($p=0.049$) (Results section: lines 251-260; Discussion section: lines 362-368).

4. The authors state that ...as noted in Table 2, we identified 10 loci which demonstrate genome-wide significance ($p \leq 5 \times 10^{-8}$) or suggestive significance ($p \leq 5 \times 10^{-6}$) of association with ROP severity, defined by stage 3 disease or greater. It should be stated instead that the authors identified one GWS locus and 9 loci of suggestive significance.

Author response: We appreciate this comment and have revised as suggested by the Reviewer and Editor.

5. Replication of these study results in independent cohorts should be conducted in future studies.

Author response: We agree with the reviewer and have expanded this discussion throughout our revised manuscript.

Minor comments:

1. The authors should consider imputation with the TopMED dataset in future analyses to more accurately address rare variants and empower the study results.

Author response: Thank you for this comment. We did not include a rare variant analysis in this work. Given the moderate size of our cohort, we focused on common variants. A rare variant analysis is beyond the scope of work at this time, but agree with the Reviewer, will be valuable in future studies as sample size increases.

REVIEWERS' COMMENTS:

Reviewer #1 (Remarks to the Author):

Dr. Li and colleagues have done a nice job of addressing my comments regarding the GWAS and eQTL analysis.

Reviewer #2 (Remarks to the Author):

The authors have adequately addressed the comments in the revised version of the manuscript, and I have no further comments.

Reviewer #3 (Remarks to the Author):

The authors have been responsive to the critique raised and the revised manuscript is significantly improved. I have no further comments